# Green Finance, Industrial Structure Upgrading, and High-Quality Economic Development–Intermediation Model Based on the Regulatory Role of Environmental Regulation

**DOI:** 10.3390/ijerph20021420

**Published:** 2023-01-12

**Authors:** Sheng Xu, Haonan Dong

**Affiliations:** College of Economics, Ocean University of China, Qingdao 266100, China

**Keywords:** green finance, quality economic development, regulated intermediation model, industrial structure upgrading, environmental regulation

## Abstract

Green finance, as a major policy innovation under the guidance of high-quality economic development, can optimize the economic development mode and structure through green investment to enhance the rationalization and advanced level of industrial structure, and ultimately enhance the level of high-quality development of China’s economy. Based on the panel data of 30 provinces and cities in China from 2009–2019, this study examines the impact of green finance on economic high-quality development and its transmission path using the intermediary model. Based on this, a model with moderated intermediation effects is constructed to explore the mechanism of green finance on an economic high-quality path based on the intermediation path of industrial structure advancement and industrial structure rationalization and the moderated effect of environmental regulation. Finally, regional heterogeneity analysis is conducted. The study found that, firstly, green finance significantly contributes to high-quality economic development, and there is a positive mediating effect of both advanced and rationalized industrial structure. Secondly, environmental regulation positively regulates the positive impact of green finance development on industrial structure upgrading and strengthens the role of industrial structure upgrading in promoting economic quality development, as well as playing a significant positive regulating role in the direct effect of green finance on economic quality development. Finally, there are regional differences in the impact of green finance on high-quality economic development.

## 1. Introduction and Literature Review

With China committing to peak carbon dioxide emissions before 2030 and achieving carbon neutrality before 2060, the Central Economic Work Conference urged quicker steps to present an action plan that enables the peaking of emissions. Green development is the core key element of high-quality economic development, and the development of green finance is urgently needed to practice green development. China is one of the countries that has established a systematic green financial policy framework, and improving the framework of the green financial system has been elevated to the national strategic level. In 2021, the balance of green credit in China has reached 15.9 trillion and the stock of green bonds has reached 1.16 trillion. Green finance is the “lubricant” and “booster” of green development, which can guide the flow of funds and enhance the level of resource allocation to promote the development of green industries [1,2]. Green finance can further inhibit the polluting behavior of high pollution and high energy-consuming enterprises under the external constraint of environmental regulation to force them to implement industrial transformation and upgrading [3], thus speeding up the transformation and upgrading of industrial structure and helping China’s transition to high-quality economic development. Therefore, exploring the path of green finance’s role in high-quality economic development from the perspective of industrial structure upgrading and environmental regulation is a necessary path to enhance the level of high-quality economic development in China.

Research in the area of the relationship between finance and economic development started early, with Goldsmith [4] having studied the relationship since 1969, and the results to date have been relatively rich; however, there has been relatively less relevant theoretical research in the niche area of green finance. Recent studies have provided some empirical evidence [5,6], and some scholars have also put forward relevant policy recommendations [7,8,9]. This paper proposes that green finance can promote high-quality economic development based on the mediating effect of industrial structural upgrading and the regulating effect of environmental regulation. No scholars have studied this issue, but some have researched green finance and its impact on industrial structural upgrading and high-quality economic development.

In the research related to green finance, since the emergence of green finance theory in the 1980s, some scholars have conducted more research on the connotation and development necessity of green finance [10,11,12]. However, the progress of green finance research is relatively slow, especially the relationship between green finance and economic quality development which started late, and there is less research literature on the transmission path of green finance to economic quality development. In earlier studies on green finance and economic development and the relationship between them, the measurement of green finance focused on single financial instruments such as green credit and green investment [13,14], the measurement of economic development focused on economic efficiency and growth quality [15], and the impact of green finance on economic development focused on green transformation and sustainable development [16,17]. In recent years, as the topic of green finance has received increasing attention, the issue of the macroeconomic effects of green finance has become an important trend in research in this area, and the relationship between green finance and economic quality has been examined more comprehensively in a number of recent studies [18,19]. Green finance not only enables the enhancement of the quality of economic development through green financial policies in sustained growth through cost and risk-sharing functions [20], but also the achievement of win–win ecological and economic benefits and contribution to high-quality economic development by guiding efficient resource allocation and green innovation [21]. In addition, green finance can transform savings into investment, thus optimizing the economic structure and stabilizing the level of economic growth, and enhancing the level of high-quality economic development [22]; other scholars have analyzed the role of green finance in optimizing the economic structure from a supply-side perspective [23]. This shows that scholars are increasingly inclined to believe that green finance does have an important function in promoting green and high-quality economic development.

In summary, on the one hand, finance can promote economic development. However, in the field of green finance, although some studies have been conducted to explore the relationship between green finance and economic development, there is still insufficient research on the mechanism of action and transmission mechanism of the relationship between green finance and high-quality economic development. In addition, there is no literature specifically based on the intermediary effect of industrial structure upgrading and the regulatory effect of environmental regulation to study the effect of green finance on economic development. On the other hand, most existing studies focus on the impact of green finance on economic aggregates, with insufficient research on regional heterogeneity. Based on this, the incremental effect of this study is reflected in: first, constructing a comprehensive index system through multiple dimensions, comprehensively examining and evaluating the level of green finance and high-quality economic development in China, and enriching and improving the definition measurement and theoretical research system of both; second, constructing a model with regulated intermediary effects, and exploring the transmission path of green finance to promote high-quality economic development based on the intermediary path of industrial structure upgrading and the regulation effect of environmental regulation; third, the total effect of green finance on high-quality economic development, with the direct effect and the indirect effect through the rationalization and upgrading of industrial structure analyzed under different levels of environmental regulation, and the moderating effect of environmental regulation explored in depth. Finally, we analyze the regional heterogeneity of the promotion effect of green finance to explore the regional development differences in China, with the findings providing a basis for promoting high-quality economic development with green finance.

## 2. Theoretical Analysis and Research Hypothesis

### 2.1. The Direct Effect of Green Finance on High-Quality Economic Development 

Green finance provides a financing path for green development and is an important tool for achieving high-quality economic development. Cowan points out that green finance can be regarded as a special service industry, which can efficiently promote green economic development through financial support strategies [2]. Green finance promotes high-quality economic development through capital formation, resource allocation, information transmission, and factor guidance functions. Specific analysis is as follows:

First, the capital formation function. Green finance allocates a variety of innovative financial instruments and their combinations to facilitate the transformation of savings into investment. This will enable the green industry to obtain sufficient funds to expand the scale of production and innovative green technology development, furthering the process of economic sustainability. Second, the resource allocation function. Green finance reduces the amount of credit financing for traditional polluting industries, forcing them to introduce green labor and production technology innovation, promoting green transformation and upgrading of industries, and promoting green development. Third, the information transfer function. The implementation of green financial policies can transmit the implementation of the country’s commitment to sustainable economic development, which in turn can change and optimize corporate behavior, innovative green technology upgrades, and production efficiency, thereby promoting the green development of the economy. Fourth, the factor guidance function. Green finance can guide the flow of funds through the role of market mechanisms, through interest rate policies, subsidy mechanisms, differential loan amounts, and other policies to reduce the risk of environmentally friendly projects, promote the establishment of green projects and the development of green industries, the formation of economies of scale, and ultimately achieve high-quality economic development. Based on this, this study proposes Hypothesis 1.

**Hypothesis** **1.**
*Green finance has a direct role in promoting high-quality economic development.*


### 2.2. Indirect Effects of Green Finance on High-Quality Economic Development through Industrial Structure Upgrading

Green finance is a key policy to achieving low-carbon economic development, which is of great significance to improve environmental quality, optimize the economic development model, and adjust the economic structure.

First, green finance concentrates capital in green industries. Green finance can pool capital through financial intermediaries and financial markets via various financing mechanisms, and guide the flow of capital to green industries such as energy conservation and emission reduction, ecological maintenance, resource conservation, and recycling through green financial tools and services, as well as differentiated credit policies, to meet the capital needs of enterprises with market competitiveness, and support the creation, production and operation, investment, and technological research and development of enterprises. At the same time, it raises the investment and financing costs of the highly polluting and energy-consuming enterprises or a series of projects, inhibits their development, and overall promotes the development of the industry to environmental protection, both rational and advanced. This is especially so with the efficient allocation of resources and the transition of the industrial structure to technology-intensive, with the highly polluting and energy-consuming enterprises gradually starting to adjust to low energy consumption and low emissions. The ecological model of green finance has been adjusted.

Second, green finance has the function of industrial integration. In the implementation of green concepts and financial policies, highly polluting and energy-consuming enterprises conduct research and development of green technologies due to the pressure of survival, reduce enterprise energy consumption and promote product upgrading to green transformation and upgrading, and improve enterprise production efficiency. In addition, Zhou [24] indicates that high pollution and high energy-consuming enterprises will also introduce the optimization of internal organizational structure and change the scale of enterprises, in order to develop new high-tech green industries, promote production factors and talents to gather in advanced high-tech green enterprises, and form a spatial agglomeration effect, so as to further optimize the resource structure in different regions of the country and finally promote the integration and upgrading of industrial structure. 

Based on the above analysis, this study proposes Hypothesis 2.

**Hypothesis** **2.**
*Green finance effectively promotes the upgrading of industrial structure, and then enhances high-quality economic development.*


In summary, the mechanism of green finance’s effect on high-quality economic development is shown in Figure 1. 

### 2.3. The Regulatory Effect of Environmental Regulation

Combining the “negative externality” of ecological pollution and the theory of market failure in economics, it is impossible to achieve the Pareto optimum by relying only on the market mechanism to allocate resources, but also must rely on government intervention, i.e., corresponding environmental regulation to ensure the reasonable allocation of resources. At the macro level, the current high quality of economic development requires, firstly, that economies increase financial investment in green environmental capital, including renewable energy and the circular economy and, secondly, that environmental policies are disciplined so that highly polluting and energy-intensive industries can green their production. All of these require the financial sector to actively guide and allocate funds between industries in order to facilitate the transformation and upgrading of industries. When green technology innovation achieves breakthroughs and green industries rise, benefiting from the realization of positive externalities of green industries, a positive feedback mechanism will be formed within the economy where economic and environmental benefits co-exist, thus achieving a win–win situation for both high-quality economic development and environmental protection. Based on the existing research, this study innovatively introduces environmental regulation as a moderating variable to analyze the impact of its moderating effect on the three paths, and then optimizes the external path boundary, in order to examine the level of economic quality development in a comprehensive way.

Firstly, concerning the regulatory effect of environmental regulation between green finance and quality economic development, green finance can be complemented by market regulation and mandatory government control, optimizing the previous paths of environmental regulation, promoting green transformation, and improving the quality of economic development [2]. Secondly, concerning the regulatory effect of environmental regulation in green finance and industrial structure, environmental regulation can be used to stimulate traditional high-polluting industries to actively implement environmental protection and emission reduction measures through effective pollution control, raising environmental access standards such as green credit and bonds, or to impose strict constraints on the negative externalities of polluting industries to regulate their environmental behavior and promote their industrial green transformation. Thirdly, the “Porter hypothesis” [25] states that environmental regulation is beneficial to economic development and that environmental policies promote the development of innovative green technologies, improve production efficiency, and promote economic development. As mentioned above, we propose Hypothesis 3.

**Hypothesis** **3.**
*Environmental regulation positively regulates the relationship between green finance and quality economic development.*


**Hypothesis** **4.**
*The relationship between environmental regulation positively regulates both green finance and industrial structure upgrading.*


**Hypothesis** **5.**
*The relationship between environmental regulation positively regulates industrial structure upgrading and high-quality economic development.*


We summarize the moderating effect of environmental regulation between green finance development, industrial structure upgrading, and high-quality economic development, as shown in Figure 2.

## 3. Indicator Selection and Modeling Methods

### 3.1. Selection of Indicators

#### 3.1.1. Explanatory Variables

Economic quality development level (EQUA): In the 13th Five-Year Plan in China, the concept of “green development” was introduced for the first time, and the social development goal of “Innovation, Coordination, Green, Openness, and Sharing” was established, which indicates that high-quality economic development has become an important feature of socialism in the new era.

On the basis of following the above development concept, the research idea of Cheng [26] is referred to and somewhat modified to comprehensively examine and evaluate the level of high-quality economic development in five dimensions, and measure the indicator weights based on the entropy value method to finally arrive at the comprehensive index of each province. The specific set indicators are shown in Table 1.

#### 3.1.2. Core Explanatory Variables

Green finance (GF): In 2016, the People’s Bank of China (PBOC) and others released the “Guiding Recommendations on Building a Green Financial System”, which optimized and improved the green financial system, indicating that green finance is a national strategic policy to promote the process of sustainable development through green credit, bonds, stock indexes, insurance, and carbon finance. Therefore, by analyzing the above guiding recommendations and referring to Zhou et al. [27], this study synthesizes the green finance index from five dimensions and seven indicators of green credit, green bonds, green insurance, green investment, and carbon finance based on the proportion of asset size in each financial field and the expert scoring method. From the above analysis, the national and regional green finance and economic quality development indices are plotted, as shown in Figure 3. The indicators set for green finance are shown in Table 2.

As can be seen from Figure 3, there are differences in the level of economic quality development and green financial development between the eastern, central, and western regions of China, with the development level of both in the eastern region much higher than that in the central and western regions. This finding is closely related to the advantageous geographical location of the eastern region and the priority development of the economy with more abundant capital and resources under the condition of policy inclination, but all regions show the trend of increasing development level. From a comprehensive perspective, China’s high-quality economic development and green financial development level have both increased, and the growth rate of high-quality economic development has slowed in recent years, while the growth rate of green financial development level has increased yearly in recent years, from 0.25 and 0.18 in 2009 to 0.33 and 0.34 in 2019, which coincides with the practical performance of China’s high-quality economic development and green finance.

The main reason for the gap between China’s high-quality economic development and the growth rate of green finance is that since the reform and the country opening up, China’s economy has continued to grow at a high rate. However, in recent years, China’s society has transitioned from a production-based society to a consumption-based society, and the goal of economic development has gradually changed from high-speed development to high-quality development, with the slowdown of China’s economic growth rate in line with the basic laws of economic development. The 2007 release of “Opinions on Implementing Environmental Protection Policies and Regulations to Prevent Credit Risks” was the beginning of the development of green finance in China. Based on the continuous development and efficiency improvement of financial institutions and markets, China has a huge potential to develop green finance and has made a series of achievements: in 2021, green credit in China accounted for 10% of the domestic loan balance, the system of green finance is tending to be perfected, the scale of products and instruments is continuing to rise, and the results of international cooperation are becoming more and more abundant. As a result, the level of green finance development has grown significantly in recent years, both nationally and in all regions.

#### 3.1.3. Mediating Variables

Industrial structure upgrading: The difference of industrial structure affects the pattern and process of economic development, and is an important discerning feature of high-quality economic development. Along with the development of green finance, China’s industrial structure has been undergoing transformation. In order to concretely show the current situation of and the level of industrial structure upgrading in China’s economy, this study divides industrial structure upgrading into two levels, industrial structure rationalization (TL) and industrial structure advancement (TS), based on the structuralist viewpoint [28].

The rationalization of industrial structure (TL) reflects the level of coordination among industries, the degree of industrial structure agglomeration, and the rationality and efficiency of production factor allocation. Referring to Gan et al. [29], the index is calculated, as shown in Equation (1).
(1)$$\mathrm{TL}=\overset{n}{\underset{i=1}{\sum }}(\frac{Y_{i}}{Y})ln\left(\frac{Y_{i}}{L_{i}}/\frac{Y}{L}\right)=\overset{n}{\underset{i=1}{\sum }}(\frac{Y_{i}}{Y})ln\left(\frac{Y_{i}}{Y}/\frac{L_{i}}{L}\right)$$

In Equation (1), $Y{\;}_{i}$ and $L_{i}$ represent the output value and the number of employed people, respectively. The value of the TL index is inversely proportional to the rationalization level of industrial structure and the TL index is positively normalized through standardization in this study, that is, the larger the TL value, the higher the rationalization level of industrial structure.

The advanced level of industrial structure (TS): In the process of the economy tending toward service, in the background that the growth rate of the tertiary industry is faster than that of the secondary industry, and the society gradually transitions from production-oriented to capital-intensive and then from capital-intensive to technology-intensive, this study uses the ratio of the output value of tertiary to secondary industry as a measure.

#### 3.1.4. Moderating Variables

Environmental regulation intensity (ER): Environmental regulation intensity can limit the polluting behavior of enterprises and influence economic development through the rational allocation of resources and enhancement of enterprise innovation capacity. In the measurement of environmental regulation indicators, scholars have put forward different views. These include Ren [29], Shen [30], ShangGuan [31], and other scholars, who proposed the category of emission reduction performance (wastewater, waste gas, and solid waste emissions, etc.), and Zhang [32], Du [33], and other scholars, who proposed the governmental actions category, while the number of non-compliant enterprises investigated and punished in each province was proposed by Chen [34] and environmental governance policies were proposed by scholars such as Li [35]. However, the above-mentioned indicators are selected with singularity, so we refer to Fu et al. [36], who selected wastewater, sulfur dioxide, and solid waste emission rates to establish a comprehensive environmental regulation indicator to measure the intensity of environmental regulation.

Firstly, the standardized value of ${\mathrm{ER}}_{ij}$ is obtained by standardizing the indicators. Secondly, the adjustment coefficient of indicators $\omega_{ij}$ is calculated to reflect the change of the ecological management intensity of each province by adjusting the weights of the three indicators.$\;\omega_{ij}$ is calculated by Equation (2):(2)$$\omega_{ij}=\left(\frac{{\mathrm{ER}}_{ij}}{\sum {\mathrm{ER}}_{ij}}\right)/\left(\frac{Y_{i}}{\sum Y_{i}}\right)$$

In Equation (2), $\omega_{ij}$ is the adjustment factor of waste *j* in province *i*; ${\mathrm{ER}}_{ij}$ and $\sum {\mathrm{ER}}_{ij}$ are the emissions of waste *j* in province *i* for the whole country; $\mathrm{and}\;Y_{i}$ and $\sum Y_{i}$ are the industrial values added in province *i* and the whole country, respectively. After obtaining $\omega_{ij}$ for each waste in the calendar year, the average value of $\omega_{ij}$ for 2009–2019 $\omega_{ij}$ is then calculated.

Finally, based on the standardized values and average weights of each indicator, the environmental regulation intensity of each province is obtained in Equation (3) as:(3)$${\mathrm{ER}}_{i}=\frac{1}{3}\sum_{j=1}^{3}\bar{\omega_{ij}}{\mathrm{ER}}_{ij}^{s}$$

#### 3.1.5. Control Variables 

There are many factors that affect the level of high-quality economic development, with the main factors controlled with reference to existing studies. The control variables selected in this study are as follows: (1) Trade dependence (TRA): The ratio of total regional import and export to total regional GDP. (2) Size of foreign investment (FI): The ratio of foreign investment amount to local GDP. (3) Human capital level (HC): The rate of general higher education graduates in this study. (4) Information level (Infor): The ratio of the number of Internet users in this study.

The above data were obtained from China Statistical Yearbook, China Science and Technology Statistical Yearbook, WIND database, CSMAR database, and provincial statistical yearbooks from 2009–2019. In the process of data acquisition, the missing data were estimated by the exponential smoothing method. The descriptive statistics of the above variables are shown in Table 3.

### 3.2. Model Construction

#### 3.2.1. Panel Model

This study takes high-quality economic development as the explained variable and green finance as the explanatory variable. To specifically analyze the impact of green finance on the qualitative development of the economy, this study conducts an empirical analysis based on panel data for 30 Chinese provinces from 2009–2019. The panel model is able to overcome the time series analysis being plagued by multiple cointegration and is able to provide more information, more variation, less cointegration, more degrees of freedom, and higher estimation efficiency. The constructed panel model (1) is built as follows:(4)$${\mathrm{EQUA}}_{it}=\alpha_{0}+\alpha_{1}{\mathrm{GF}}_{it}+\sum \alpha_{i}control_{it}+\varepsilon_{it}$$

In the model, *i* = 1, 2, 3, …, *N* represents each province; *t* = 1, 2, 3, …, *N* represents each year; ${\mathrm{EQUA}}_{it}$ is the level of high-quality economic development; ${\mathrm{GF}}_{it}$ is the level of green finance; $control_{it}\;$ is the control variable; $\mathrm{and}\;\varepsilon_{it}$ is the random disturbance term.

#### 3.2.2. Intermediary Effect Model

Based on the analysis of the impact of green finance on economic quality development, we then explore whether green finance can have an impact on the level of economic quality development through industrial structure upgrading. Therefore, referring to Wen [37], we establish the following model (2) and (3) for the intermediary effect test.
(5)$${\mathrm{TL}}_{it}\left({\mathrm{TS}}_{it}\right)=\beta_{0}+\beta_{1}{\mathrm{GF}}_{it}+\sum \beta_{i}control_{it}+\varepsilon_{it}$$
(6)$${\mathrm{EQUA}}_{it}=\gamma_{0}+\gamma_{1}{\mathrm{GF}}_{it}+\gamma_{2}{\mathrm{TL}}_{it}\left({\mathrm{TS}}_{it}\right)+\sum \gamma_{i}control_{it}+\varepsilon_{it}$$

In the model, *i* and *t* denote provinces and years, respectively, with the above analysis dividing industrial structure upgrading into industrial structure rationalization (i.e., ${\mathrm{TL}}_{it}$) and industrial structure advanced (i.e., ${\mathrm{TS}}_{it})$. Among the coefficients, $\beta_{1}\gamma_{2}$ is the mediating effect and $\gamma_{1}$ is the direct effect, in which the proportion of the mediating effect to the total effect is $\beta_{1}\gamma_{2}/\alpha_{1}$.

#### 3.2.3. Mediating Effect Model with Regulation

To further test the moderating effect of environmental regulation on the mediating effect, referring to Wen [38], based on the previous analysis of the moderating mechanism of environmental regulation, the moderated mediating model Equations (4), (5), and (6) are constructed as follows:(7)$${\mathrm{EQUA}}_{it}=\alpha_{0}+\alpha_{1}{\mathrm{GF}}_{it}+\alpha_{2}{\mathrm{ER}}_{it}+\alpha_{3}\left({\mathrm{GF}}_{it}{*\mathrm{ER}}_{it}\right)+\sum \alpha_{i}control_{it}+\varepsilon_{it}$$
(8)$$\mathrm{TL}{\left(\mathrm{TS}\right)}_{it}=\beta_{0}+\beta_{1}{\mathrm{GF}}_{it}+\beta_{2}{\mathrm{ER}}_{it}+\beta_{3}\left({\mathrm{GF}}_{it}{*\mathrm{ER}}_{it}\right)+\sum \beta_{i}control_{it}+\varepsilon_{it}$$
(9)$$\;{\mathrm{EQUA}}_{it}=\alpha_{0}+\alpha_{1}^{{}^{\prime }}{\mathrm{GF}}_{it}+\alpha_{2}^{{}^{\prime }}{\mathrm{ER}}_{it}+\alpha_{3}^{{}^{\prime }}({\mathrm{GF}}_{it}{*\mathrm{ER}}_{it})+\gamma_{1}{\mathrm{TL}}_{it}\left({\mathrm{TS}}_{it}\right)+\gamma_{2}\left({\mathrm{TL}}_{it}\left({\mathrm{TS}}_{it}\right){*\mathrm{ER}}_{it}\right)+\sum a_{i}control_{it}+\varepsilon_{it}$$
where ${\mathrm{ER}}_{it}$ denotes environmental regulation. In this study, the regulating effect of environmental regulation is tested by the stepwise regression method: first, in the regression of model (7), if the coefficient $\alpha_{3}$ is significant, it means that the regulating effect of environmental regulation on the relationship between green finance and economic quality development is significant; second, in the regression of models (8) and (9), the significance of $\beta_{1}$ and $\beta_{3}$ in model (8), then the significance of $\gamma_{1}$ and $\gamma_{2}$ in model (9), is tested. If the regression coefficients of $\beta_{1}\gamma_{2}$, $\beta_{3}\gamma_{1}$, and $\alpha_{3}\gamma_{2}$ are non-zero, so long as one of the three groups is not zero, it is known that there is a moderating effect of environmental regulation. Under the moderating effect of environmental regulation, the total effect of green finance on high-quality economic development is $\alpha_{1}$ + $\alpha_{3}{\mathrm{ER}}_{it}$ and the effect on industrial structure upgrading is $\beta_{1}$ + $\beta_{3}{\mathrm{ER}}_{it},$ the mediating effect of industrial structure upgrading is ($\beta_{1}+\beta_{3}{\mathrm{ER}}_{it}$)($\;\gamma_{1}+\gamma_{2}{\mathrm{ER}}_{it}$), and $\alpha_{1}^{{}^{\prime }}+\alpha_{3}^{{}^{\prime }}{\mathrm{ER}}_{it}$ denotes the direct effect of green finance on high-quality economic development under the moderating effect of environmental regulation.

## 4. Empirical Analysis of Green Finance for High-Quality Economic Development

### 4.1. Regression of Green Finance on High-Quality Economic Development

The regression results of green finance on high-quality economic development are shown in Table 4. The basic equation is an empirical test with the introduction of only the core explanatory variable green finance; the extended equation is an empirical test with the gradual introduction of the control variables human capital level, informationization level, trade dependence, and foreign investment scale. In the regression results of model (4), the coefficients of green finance are all significantly positive at the 1% level, indicating the positive contribution of green finance to high-quality economic development. In the extended equation, human capital level, informationization level, foreign trade dependence, and foreign investment scale are all significantly positive, which is in line with the expectation of this study.

The reason: green finance helps to share information among enterprises, government and financial institutions, which enables them to enhance the efficiency of capital factors’ utilization and rationalization of resource allocation. In addition, with the effective regulation of the market mechanism, green enterprises have multiple paths to obtain credit funds, achieve efficient allocation of production factors, and gradually realize economies of scale, which will strongly improve the development of green finance and realize its effective integration with economic development to improve the quality of economic growth. Therefore, the development of green finance has a positive effect on the improvement of economic growth rate, which is consistent with H1.

### 4.2. The Intermediary Effect Test of Industrial Structure Upgrading

#### 4.2.1. Intermediary Effect Test

In the basic regression analysis, the coefficients of green finance are significantly positive, so the intermediation effect test can be conducted. The regression results of the mediating effects of industrial structure upgrading are shown in Table 5. In model (5), the coefficients of green finance are 0.493 and 2.293, respectively, which are significantly positive, that is, there is a significant positive relationship between green finance and industrial structure upgrading. In model (6), the coefficients of green finance are 0.295 and 0.310, respectively, and the coefficients of industrial structure rationalization and upgrading are 0.010 and 0.017, respectively, both of which are significantly positive, indicating that the mediating effect of implementing industrial structure upgrading is significant. To sum up, in addition to the direct effect of green finance on economic quality development, it can also achieve economic quality development by promoting industrial structure upgrading, i.e., there is a partial intermediary effect. According to the calculation of the coefficients of each parameter, the indirect effects of green finance on economic quality development through the rationalization and upgrading of industrial structure are 0.054 and 0.039, respectively, and the total effect is 0.349, while the intermediary effects are 16% and 11%, respectively, and the influence direction is positive, and H2 is verified.

The reasons: First, for industrial structure upgrading, green finance can inhibit the capital inflow of polluting industries; promote the factor gathering of green industries; promote the gradual transformation of industrial structure from labor-intensive to capital-intensive, and then from capital-intensive to technology-intensive; enhance the degree of coordination among industries; and improve the industrial production through the functions of capital orientation, resource allocation, information transfer, and factor integration. It enhances the coordination between industries, improves the allocation efficiency of industrial production elements and resources, and raises the level of industrial rationalization and industrial advancement. On this basis, green finance has an impact on economic growth in terms of both structure and efficiency: in terms of industrial structure, due to the continuous growth of green finance and the gradual expansion of the share of green industries in China, the highly polluting and energy-consuming enterprises are shrinking, and the industrial structure shows the development trend of ecological greening, thus driving the high-quality growth of China’s economy; in the area of comprehensive efficiency, under the comprehensive influence of the risk mechanism, all types of enterprises have been able to improve their competitiveness. In the area of comprehensive efficiency, under the comprehensive influence of risk mechanism, all types of enterprises take the initiative to implement green technology innovation, thus further optimizing the capital efficiency and improving the quality of economic growth.

#### 4.2.2. Sobel–Goodman and Bootstrap Tests

In order to enhance the robustness of the mediation effect test, both Sobel–Goodman and Bootstrap tests are used to validate this study. Table 4 shows the results of the two tests to verify the rationalization and advanced industrial structure. When the Sobel–Goodman test is conducted and the results are shown in Table 6, the Z-values of industrial structure rationalization and industrial structure advancement are both greater than 1.65 and significant at the 1% level, and when the Bootstrap test is conducted and the results are shown in Table 6, the 95% confidence interval of both the indirect and direct effects do not contain zero, which shows that there is a partial mediating effect of industrial structure rationalization and advancement, which again verifies H2.

### 4.3. Moderating Effect Test

In the regression results of the mediating effect model with regulation in Table 7, the coefficients of GF∗ER in model (7) are 0.429 and 0.279 and are both significant at the 1% level, indicating that environmental regulation positively regulates the total effect of green finance on economic quality development; the regression coefficient of the interaction term GF∗ER in model (8) is significant, indicating that the positive regulating effect of environmental regulation on the direct path is significant. This verifies H3. Further, according to the coefficients 1.029 and 2.354 of the interaction term GF∗ER of model (8), the regression coefficients 0.028 and 0.025 of TL∗ER and TS∗ER of model (9) are significantly positive, so it can be concluded that environmental regulation has a significant moderating effect on the indirect path of green finance affecting economic high-quality development through industrial structure upgrading, which is consistent with H4 and H5.

### 4.4. Changes in the Impact of Green Financial Development on the Level of Economic Quality Development through Industrial Structure Upgrading under the Moderating Effect of Different Environmental Regulation Levels (Negative One Standard Deviation of the Mean, Mean, Positive One Standard Deviation of the Mean)

In this study, we use the Bootstrap method to test the mediating effect model with regulation: the mediating effect of industrial structure upgrading under the moderating effect of different environmental regulation levels, i.e., low environmental regulation level, mean environmental regulation level, and high environmental regulation level, respectively, with the results shown in Table 8.

First, for the conditional direct effect, we can see that with TL as the mediating variable, the conditional direct effects of green finance are 0.2091, 0.3554, and 0.5017 at low-to-high ER levels, respectively; with TS as the mediating variable, the conditional direct effects of green finance are 0.2403, 0.3945, and 0.5487, all gradually increasing, and all with confidence intervals not containing 0. This indicates that the direct effect of green finance on high-quality economic development increases significantly as the ER rises.

Second, for the conditional mediating effect, we can see that when TL is the mediating variable, the conditional mediating effects of TL are −0.007, 0.0628, and 0.1764, respectively, and the confidence interval does not contain 0 and the coefficient is significant only when ER is the mean and positive one standard deviation of the mean; with TS as the mediating variable, the conditional mediating effects of TS are −0.0151, 0.0208, and 0.1461, respectively, and the confidence interval does not contain 0 when ER is positive one standard deviation of the mean. The confidence interval does not contain 0 until the ER is one standard deviation, which shows that the mediation effect of TS is only highlighted when the ER is large.

Third, for the conditional total effect, we can see that the total effect of green finance on economic quality development is 0.183, 0.4273, and 0.6715 with the confidence interval not containing 0. This indicates that the promotion effect of green finance on economic quality development increases significantly with the increase of ER level.

The regression results plot the moderating effect M ± 1SD and J-N, as shown below. From Figure 4, it can be seen that the total effect of green finance on high-quality economic development increases significantly with the increasing level of environmental regulation. According to Figure 5a, when the moderating variable environmental regulation is greater than 0.5073, the positive effect of green finance on the rationalization of industrial structure is significant, and the positive effect keeps expanding as the level of environmental regulation increases. According to Figure 5b, the positive effect of green finance on the advanced industrial structure is significant at any level of environmental regulation, and still the positive effect keeps increasing with the increasing level of environmental regulation. According to Figure 6a,b, the promotion effect of industrial structure rationalization on economic high-quality development is significant under different environmental regulation; when the level of environmental regulation is greater than 0.2082, while the advanced role of industrial institutions on economic high-quality development enhancement is significant, both have an increasing positive promotion effect as the level of environmental regulation increases.

The above analysis shows that an increase in the intensity of environmental regulation is conducive to the further realization of high-quality economic development. At present, the most common environmental policy in China is directive environmental regulation, the main means of which is to force enterprises to meet the emission and technical standards set by the environmental protection authorities through relevant policies. The government should use environmental regulation to force polluting enterprises to take the path of low-carbon and green development. On the other hand, the government should guide green finance to support the research and development and promotion of new technologies and products that are green, energy-saving, and emission-reducing, in order to realize the vision of environmental regulation and green financial innovation. The government should also guide green finance to support the research, development, and promotion of new technologies and energy-saving and emission-reducing products, in order to realize the vision of environmental regulation and green financial innovation to enhance high-quality economic development.

### 4.5. Regression Test of Green Finance on Economic Quality Development in the East and the Middle and West

In this study, China is divided into East, Central, and West according to the geographical location and economic development for regional heterogeneity analysis, and a regression analysis is conducted to test the regional heterogeneity of the impact of green finance development on economic quality development based on the selection of control variables such as foreign trade dependence, foreign investment scale, and human capital. and the regression results are shown in Table 9. The empirical results show that green finance in both eastern and central and western regions of China can significantly improve the level of economic high quality at the 1% level, but there are regional differences. China is a vast country, and different regions have different economic, geographical, and ecological conditions. The eastern region of China has an advantageous geographic location, and its economy is more abundant in funds and resources under the condition of policy inclination, so it has taken the lead in developing its economy to enhance rapid growth. The development of green finance in the eastern region has reached a higher level than that in the central and western regions. This is due to the fact that the eastern region is relatively rich in green financial instruments, and enterprises mainly use direct financing such as green bonds and green funds; the market is more efficient and has been developed as a priority, so the role of green finance in promoting high-quality economic development is gradually weakened compared with the beginning. Secondly, the development level of green finance in the central and western regions is relatively backward, the economic foundation is poor, the financial instruments and services at the disposal of green finance are relatively simple, the market mechanism is not perfect, and the allocation of funds and resources is not efficient enough. Therefore, if the central and western regions increase the investment and resource allocation of green finance on the basis of the current level of green finance, then its positive impact on the high-quality economic development will be significantly enhanced. The country should give strong policy support to the central and western regions, as well as increase the investment of funds and the introduction of talents and other policies to enhance the level of its green financial development.

### 4.6. Robustness Tests

In order to ensure the robustness of the findings on the impact of green finance on economic quality development, this section conducts robustness tests through the lagged effect of green finance, panel quantile model, and sequential elimination of control variables.

#### 4.6.1. Lagged Effect of Green Finance

High-quality economic development is influenced by both green finance and changes due to the time lag of green finance development. To ensure the robustness of the above findings in this study, green finance with a lag of one period is treated as an explanatory variable and thus subjected to panel regression. As can be seen from the regressions in Table 10, the coefficient of green finance is still significantly positive, representing the good robustness of the regression results in this study.

#### 4.6.2. Panel Quantile Regression of Green Finance on Economic Quality Development

In the previous paper, the traditional least squares method was used to estimate the mean impact of green finance on economic quality development; quantile regression can reflect the overall distribution level of economic quality development, i.e., the degree of green finance impact under economic quality development in different quantiles. The results in Table 11 show that the coefficients of green finance in different quantiles are still significantly positive, which proves the robustness of the results of this study.

#### 4.6.3. Regression of Green Finance on Economic Quality Development Excluding Control Variables Test

Since the regression results and the obtained conclusions may vary depending on the control variables, this study excludes the control variables of foreign capital scale, foreign trade dependence, informationization level, and human capital level in turn and then conducts the robustness test.

The regression results are shown in Table 12. Although the coefficients of green finance are different from the regression results in Table 4, the sign and significance are consistent, which means that the above regression results have robustness and again verify the conclusions of this study.

## 5. Conclusions

With the transformation of China’s economy from high-speed development to high-quality development, which makes higher requirements for the country’s economic development model and economic structure, China urgently needs to implement green development, and green development requires vigorous development of green finance. Based on economic phenomena and the frontier literature, this study explores the intrinsic mechanism of green finance to promote high-quality economic development based on the mediating effect of industrial structure upgrading and the regulating effect of environmental regulation based on panel data of 30 provinces in China from 2009–2019. The study finds that, firstly, green finance has a significant positive direct contribution to high-quality economic development in China. Secondly, there is a positive partial mediating effect of industrial structure rationalization and industrial structure advancement between green finance and high-quality economic development. Thirdly, for the moderated mediation model, environmental regulation positively moderates the relationship between green finance, industrial structure upgrading, and high-quality economic development, and the positive relationship between the two is enhanced as the intensity of environmental regulation continues to increase. Fourthly, in the heterogeneity analysis of the regional panel regression model, there are regional differences in the impact of green finance on high-quality economic development between the eastern and central and western regions. However, in the above-mentioned process, factors such as the level of green finance development, the rationalization of local industrial structure, and the level of advanced industrial structure, as well as the intensity of environmental regulation, influence and determine the effect of green finance. Therefore, it is necessary for the state, local governments, financial institutions, and enterprises to work together to improve the green financial system in order to promote China’s high-quality economic development. Accordingly, the following policy recommendations are proposed.

First, a national coordinated design and improvement of green financial system to promote sustainable development of green finance. In recent years, the National Development and Reform Commission, the Ministry of Finance, and the Ministry of Environmental Protection have launched a series of green financial reform initiatives one after another, and have achieved certain results. However, it is difficult to achieve the goal of high-quality economic development with the existing green financial policies and fragmented measures. Therefore, the first step is to strengthen the top-level design of the green financial system and enhance the intensity of policy implementation. A unified standard system for green finance must be developed and improved in conjunction with market practice, and needs to be gradually refined, clarified, and made operable, in order to gradually expand the scope of standards for the green financial system in building a universal and unified green financial system. Innovation in green financial products and services should be encouraged to provide diversified green financial products, and the enthusiasm of market players to innovate green financial business models needs to be fully mobilized in order to effectively improve the performance of green financial business.

Second, accelerate the upgrading of industrial structure and assist high-quality economic development. Research findings show that the upgrading of industrial structure provides a key intermediary path for green finance to drive China’s high-quality economic development. On the one hand, China’s industrial structure should be optimized by vigorously developing emerging enterprises, breaking through the strategic layout of core technologies, and not making massive cuts to highly polluting and energy-consuming enterprises, instead adopting a guiding and incentivized approach to promote them to complete their green transformation. On the other hand, China should enhance the suitability of green financial and fiscal policies, minimize the cyclical fluctuations of policy effects, and ensure the long-term sustainability of policy effects, in order to provide stable financial subsidies and preferential policies for enterprises’ green transformation.

Third, optimize the environmental regulation policy channel and enhance the efficiency of environmental policy implementation. China should flexibly exert the influence of environmental regulation on green finance and industrial structure upgrading, and adopt an approach that encourages and regulates both according to the characteristics of different industries. Each region should adopt reasonable environmental regulation measures, according to the characteristics of their respective industrial structures and resource endowment conditions, in order to give full play to the reversing effect of environmental regulation on industrial structure optimization and promote regional industrial transformation and high-quality economic development.

Finally, adhere to local conditions and implement differentiated regional policies. China is a vast country, and the level of green finance and high-quality economic development in different regions exhibits regional heterogeneity due to economic, geographical, cultural, and policy conditions. The eastern region should adhere to and improve the existing green financial system, vigorously promote innovation in the green financial system, and continue to maintain the determination for green and high-quality development. For the central and western regions, efforts such as fiscal transfer payments should be increased to find regional advantages, combine regional resource advantages with green finance, achieve differentiation in positioning, and take a leapfrog development path to further narrow the regional gap.

## Figures and Tables

**Figure 1 ijerph-20-01420-f001:**
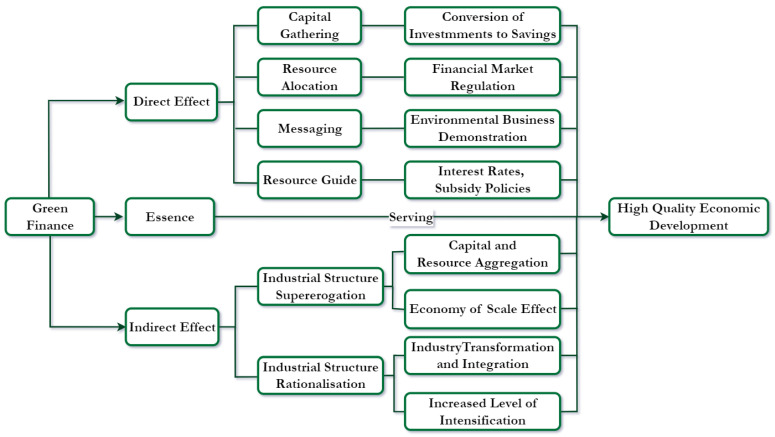
Mechanism of action of green finance and high-quality economic development.

**Figure 2 ijerph-20-01420-f002:**
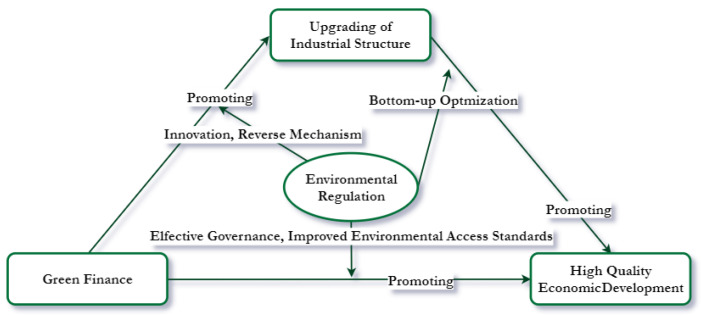
Mechanism of the regulatory role of environmental regulation.

**Figure 3 ijerph-20-01420-f003:**
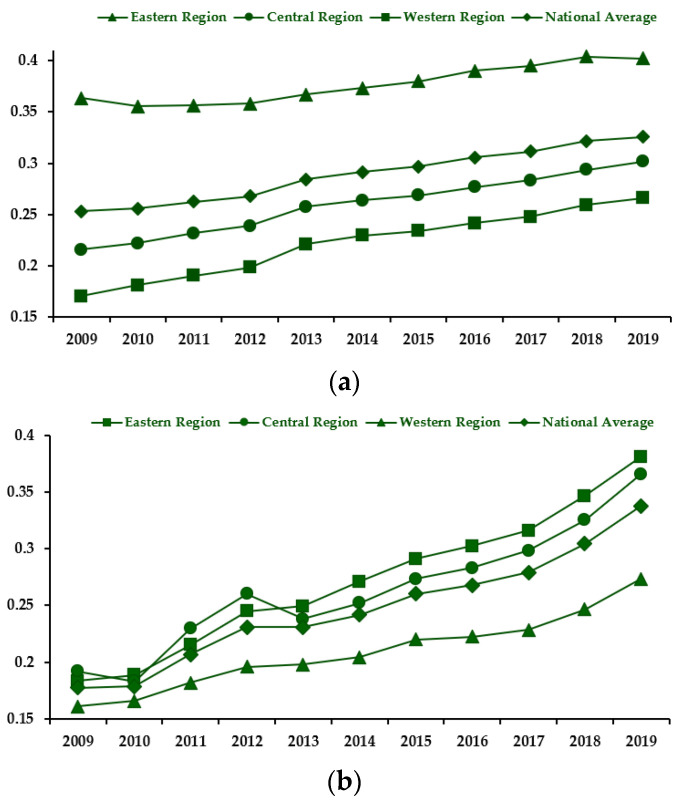
(**a**) National and regional levels of high-quality economic development. (**b**) National and regional levels of green financial development.

**Figure 4 ijerph-20-01420-f004:**
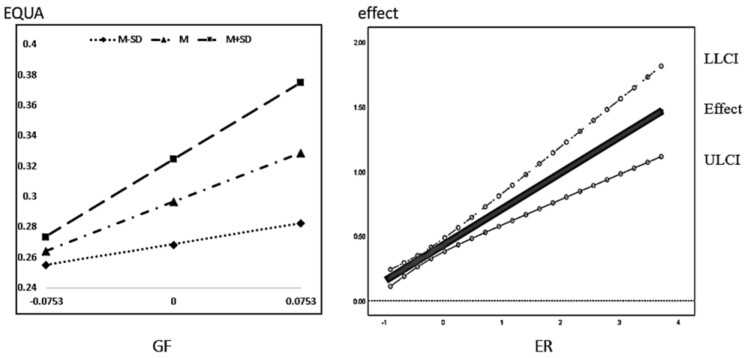
Total effect of green finance on economic quality development under the role of environmental regulation.

**Figure 5 ijerph-20-01420-f005:**
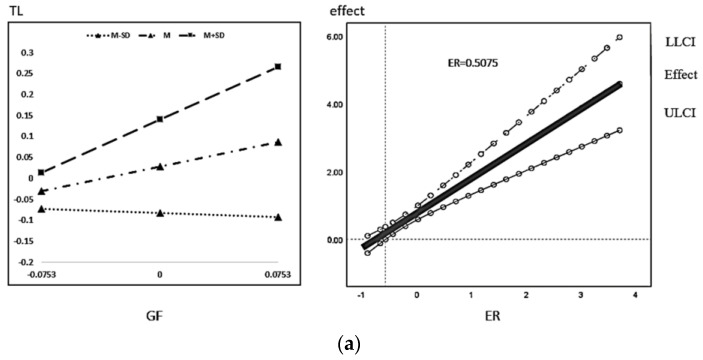
(**a**). Impact of green finance on the rationalization of industrial structure under the role of environmental regulation. (**b**) Impact of green finance on the advanced industrial structure under the role of environmental regulation.

**Figure 6 ijerph-20-01420-f006:**
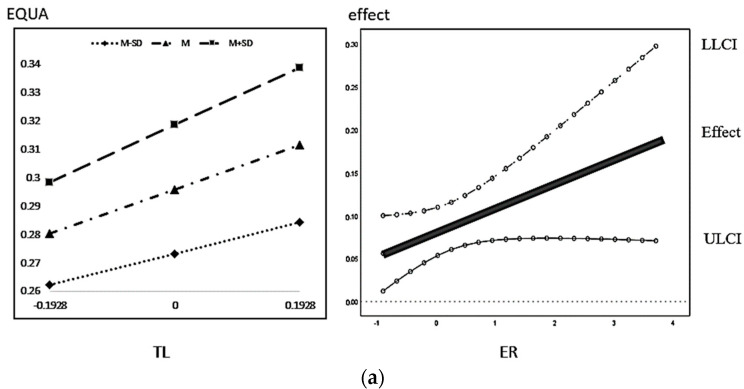
(**a**) Impact of industrial structure rationalization regulated by environmental regulation on economic quality development. (**b**) Impact of advanced industrial structure regulated by environmental regulation on high-quality economic development.

**Table 1 ijerph-20-01420-t001:** Economic quality development index system.

Level 1 Indicators	Level 2 Indicators	Level 4 Indicators	Indicators Meaning	Indicator Properties
Marine Economy Quality Development Index	Innovation	Researcher input ratio	Number of scientific researchers/Year-end resident population	(+)
		Investment in scientific research	Scientific research expenditure/Local GDP	(+)
		Technology market turnover share	Technology market turnover/Local GDP	(+)
	Coordination	Comparison of urban and rural income	Per capita disposable income of urban residents/Per capita disposable income of rural residents	(−)
		Comparison of urban and rural consumption	Per capita consumption expenditure of urban residents/Per capita consumption expenditure of rural residents	(−)
		Comparison of economic development levels	GDP per capita by province/GDP per capita nationwide	(+)
	Green	Energy consumption level	Total energy consumption/Local GDP	(−)
		Forest cover	Forest cover	(+)
		Exhaust emissions	Emissions/Local GDP	(−)
		Solid waste emissions	Solid waste emissions/Local GDP	(−)
	Openness	Percentage of import and export trade volume	Total imports and exports/GDP	(+)
		Percentage of foreign investment	Amount of foreign investment/GDP	(+)
		Share of international visitors received	Number of international visitors received/Year-end resident population	(+)
	Sharing	Education development level	General university student-teacher ratio	(−)
		Public library collections per capita	Public library holdings per capita	(+)
		Number of medical facility beds per capita	Number of medical facility beds per capita	(+)

**Table 2 ijerph-20-01420-t002:** Green finance indicator system.

Level 1 Indicators	Level 2 Indicators	Level 3 Indicators	Indicators Meaning	Indicator Properties
Green Finance Index	Green Credit	Percentage of green credit	Green credit loan amount/Total loan amount	(+)
		Percentage of interest expenses in six energy-consuming industries	Six major energy-consuming industries Interest expenses/Total interest of industrial industries	(−)
	Green Securities	Percentage of total market capitalization of environmental companies	Total market capitalization of environmental companies/Total A-share market capitalization	(+)
		Six high energy-consuming industries accounted for by market value	Market capitalization/Total market capitalization of the six major energy-consuming industries	(−)
	Green Insurance	Share of agricultural insurance expenditures	Agricultural insurance noted/Total insurance expenses	(+)
	Green Investment	Energy saving and environmental protection investment ratio	Energy saving and environmental protection investment/Fiscal expenditure	(+)
	Carbon Finance	CO_2_ emissions as a percentage	CO_2_ emissions/Local GDP	(−)

**Table 3 ijerph-20-01420-t003:** Descriptive statistics of the variables.

Variable	N	Mean	P50	SD	MAX	MIN
EQUA	330	0.289	0.276	0.105	0.597	0.103
GF	330	0.247	0.232	0.075	0.551	0.139
TS	330	1.123	0.939	0.645	5.154	0.499
TL	330	0.222	0.203	0.136	0.702	0.0161
ER	330	1.061	0.790	0.875	4.768	0.162
HC	330	0.254	0.256	0.023	0.299	0.000
TRA	330	0.276	0.135	0.317	1.548	0.013
FI	330	0.372	0.203	0.399	3.730	0.048
INFOR	330	0.056	0.04	0.039	0.236	0.014

**Table 4 ijerph-20-01420-t004:** Basic regression results of green finance on the quality development of the economy.

Variables	Model (1)				
	Basic Equation	Extended Equations			
GF	0.359 ***	0.353 ***	0.334 ***	0.351 ***	0.349 ***
	(0.021)	(0.020)	(0.021)	(0.023)	(0.023)
HC		24.022 ***	26.687 ***	26.161 ***	26.218 ***
		(6.060)	(6.051)	(6.023)	(6.004)
INFOR			0.094 ***	0.093 ***	0.076 **
			(0.032)	(0.032)	(0.033)
TRA				0.031 **	0.040 **
				(0.015)	(0.016)
FI					0.011 *
					(0.006)
_cons	0.200 ***	0.141 ***	0.133 ***	0.122 ***	0.117 ***
	(0.005)	(0.016)	(0.016)	(0.017)	(0.017)
N	330.000	330.000	330.000	330.000	330.000
R^2^	0.502	0.527	0.541	0.547	0.552
R^2^_a	0.452	0.478	0.491	0.497	0.500

Note: Standard deviations are in parentheses, where ***, **, and * indicate significance at the 1%, 5%, and 10% levels, respectively.

**Table 5 ijerph-20-01420-t005:** Intermediary model regression results.

Variables	Model (2)		Model (3)	
	TL	TS	EUQA	EQUA
GF	0.493 ***	2.293 ***	0.295 ***	0.310 ***
	(0.088)	(0.213)	(0.022)	(0.026)
TL			0.110 ***	
			(0.013)	
TS				0.017 ***
				(0.006)
HC	44.165 *	66.567	21.377 ***	25.078 ***
	(23.454)	(56.649)	(5.468)	(5.949)
INFOR	0.407 ***	1.660 ***	0.032	0.048
	(0.129)	(0.311)	(0.030)	(0.034)
TRA	0.064	−0.986 ***	0.033 **	0.056 ***
	(0.062)	(0.149)	(0.014)	(0.017)
FI	−0.023	0.054	0.013 **	0.010
	(0.024)	(0.058)	(0.006)	(0.006)
_cons	0.459 ***	0.547 ***	0.067 ***	0.108 ***
	(0.066)	(0.160)	(0.017)	(0.017)
N	330.000	330.000	330.000	330.000
R^2^	0.172	0.568	0.634	0.563
R^2^_a	0.077	0.519	0.590	0.511

Note: Standard deviations are in parentheses, where ***, **, and * indicate significance at the 1%, 5%, and 10% levels, respectively.

**Table 6 ijerph-20-01420-t006:** Sobel–Goodman and Bootstrap tests.

	Sobel–Goodman	Z	*p*	Bootstrap	*p*	LLCI	ULCI
TL	Sobel Goodman-1 Goodman-2	4.604 *** 4.58 *** 4.628 ***	4.150 × 10^−6^ 4.644 × 10^−6^ 3.699 × 10^−6^	Ind_eff Dir_eff	0.004 0.000	0.0175842 0.2282403	0.0905101 0.3614135
TS	Sobel Goodman-1 Goodman-2	2.716 *** 2.705 *** 2.727 ***	0.0066106 0.0068318 0.0063933	Ind_eff Dir_eff	0.050 0.000	0.0000385 0.2294834	0.0784947 0.3897315

Note: Standard deviations are in parentheses, where *** indicate significance at the 1% levels, respectively.

**Table 7 ijerph-20-01420-t007:** Regression results of the mediated effects model with adjustment.

Variables	Model (4)	Model (5)		Model (6)	
	EUQA	TL	TS	EQUA	
GF	0.427 ***	0.777 ***	3.068 ***	0.355 ***	0.395 ***
	(0.026)	(0.102)	(0.251)	(0.027)	(0.031)
TL				0.081 ***	
				(0.014)	
TS					0.007
					(0.006)
ER	0.032 ***	0.127 ***	0.058	0.026 ***	0.040 ***
	(0.010)	(0.038)	(0.093)	(0.010)	(0.010)
GF∗ER	0.279 ***	1.029 ***	2.354 ***	0.167 ***	0.176 ***
	(0.044)	(0.171)	(0.422)	(0.045)	(0.050)
TL∗ER				0.028 *	
				(0.016)	
TS∗ER					0.025 ***
					(0.006)
Control Variables	Control	Control	Control	Control	Control
_cons	0.084 ***	0.331 ***	0.441 **	0.057 ***	0.082 ***
	(0.018)	(0.069)	(0.171)	(0.017)	(0.017)
N	330.000	330.000	330.000	330.000	330.000
R^2^	0.614	0.279	0.610	0.662	0.635
R^2^_a	0.567	0.190	0.562	0.618	0.588

Note: Standard deviations are in parentheses, where ***, **, and * indicate significance at the 1%, 5%, and 10% levels, respectively.

**Table 8 ijerph-20-01420-t008:** Test results of direct, mediated, and total effect paths of environmental regulation.

	ER	Effect	Boot SE	LLCI	ULCI
Conditional Direct Effects (TL as a mediating variable)	−0.8753 (M − SD)	0.2091	0.0315	0.147	0.2711
	0 (M)	0.3554	0.0268	0.3026	0.4082
	0.8753 (M + SD)	0.5017	0.06	0.3837	0.6197
Conditional Direct Effects (TS as a mediating variable)	−0.8753 (M − SD)	0.2403	0.0359	0.1697	0.3109
	0 (M)	0.3945	0.0313	0.3329	0.4561
	0.8753 (M + SD)	0.5487	0.0672	0.4164	0.681
Conditional Mediation Effect (TL as a mediating variable)	−0.8753 (M − SD)	−0.007	0.0085	−0.0259	0.0075
	0 (M)	0.0628	0.0259	0.025	0.1264
	0.8753 (M + SD)	0.1764	0.0654	0.0646	0.3135
Conditional Mediation Effect (TS as a mediating variable)	−0.8753 (M − SD)	−0.0151	0.0169	−0.0551	0.0117
	0 (M)	0.0208	0.0288	−0.0367	0.0767
	0.8753 (M + SD)	0.1461	0.0476	0.0512	0.2394
Total Condition Effect	−0.8753 (M − SD)	0.183	0.0322	0.1197	0.2463
	0 (M)	0.4273	0.026	0.3762	0.4783
	0.8753 (M + SD)	0.6715	0.0568	0.5597	0.7832

Note: LLCI = lower limit of 95% confidence interval, ULCI = upper limit of 95% confidence interval.

**Table 9 ijerph-20-01420-t009:** Regional empirical tests of green finance for quality economic development.

Variables	Model (1)	Model (2)	Model (3)
	Eastern Region	Central Region	Western Region
GF	0.177 ***	0.320 ***	0.633 ***
	(0.029)	(0.036)	(0.050)
HC	31.983 **	18.380 ***	37.230 ***
	(13.569)	(6.653)	(12.482)
INFOR	0.168 ***	−0.153 *	−0.009
	(0.060)	(0.080)	(0.046)
TRA	−0.022	−0.102	0.022
	(0.015)	(0.094)	(0.049)
FI	0.002	0.186 ***	0.069 **
	(0.005)	(0.052)	(0.032)
_cons	0.247 ***	0.111 ***	−0.014
	(0.038)	(0.021)	(0.030)
N	121.000	88.000	121.000
R^2^	0.563	0.674	0.723
R^2^_a	0.500	0.622	0.684

Note: Standard deviations are in parentheses, where ***, **, and * indicate significance at the 1%, 5%, and 10% levels, respectively.

**Table 10 ijerph-20-01420-t010:** Lagged effects of green finance.

Variables	Model (1)	Model (2)	Model (3)	Model (4)	Model (5)
	Basic Equation	Extended Equations			
L.GF	0.382 ***	0.375 ***	0.353 ***	0.379 ***	0.376 ***
	(0.022)	(0.022)	(0.022)	(0.024)	(0.024)
HC		22.745 ***	26.577 ***	25.817 ***	25.877 ***
		(6.132)	(6.006)	(5.948)	(5.931)
INFOR			0.138 ***	0.139 ***	0.123 ***
			(0.031)	(0.030)	(0.032)
TRA				0.041 ***	0.049 ***
				(0.015)	(0.016)
FI					0.010 *
					(0.006)
_cons	0.201 ***	0.145 ***	0.132 ***	0.117 ***	0.112 ***
	(0.005)	(0.016)	(0.016)	(0.017)	(0.017)
N	330.000	330.000	330.000	330.000	330.000
R^2^	0.494	0.517	0.547	0.558	0.563
R^2^_a	0.444	0.466	0.499	0.509	0.512

Note: Standard deviations are in parentheses, where ***, and * indicate significance at the 1%, and 10% levels, respectively.

**Table 11 ijerph-20-01420-t011:** Panel quantile regressions of green finance on quality economic development.

Variables	Model (1)	Model (2)	Model (3)	Model (4)	Model (5)
	10%	25%	50%	75%	90%
GF	0.341 ***	0.261 ***	0.210 ***	0.504 ***	0.518 ***
	(0.048)	(0.055)	(0.077)	(0.030)	(0.052)
FI	0.050 ***	0.113 ***	0.077 ***	0.063 ***	0.054 ***
	(0.011)	(0.013)	(0.018)	(0.007)	(0.012)
TRA	0.158 ***	0.158 ***	0.196 ***	0.221 ***	0.247 ***
	(0.014)	(0.016)	(0.023)	(0.009)	(0.015)
INTER	−0.184 **	−0.191 *	−0.143	−0.054	−0.046
	(0.091)	(0.104)	(0.147)	(0.057)	(0.100)
HC	23.666	11.475	46.518 *	36.627 ***	50.019 ***
	(15.837)	(18.094)	(25.616)	(9.856)	(17.323)
_cons	0.024	0.079 *	0.050	0.032	0.010
	(0.040)	(0.046)	(0.065)	(0.025)	(0.044)
N R^2^_a	330.000 0.420	330.000 0.4330	330.000 0.4561	330.000 0.5872	330.000 0.6839

Note: Standard deviations are in parentheses, where ***, **, and * indicate significance at the 1%, 5%, and 10% levels, respectively.

**Table 12 ijerph-20-01420-t012:** Green finance on high-quality economic development regression excluding control variables test.

Variables	Model (1)	Model (2)	Model (3)	Model (4)
	EQUA			
GF	0.351 ***	0.330 ***	0.363 ***	0.361 ***
	(0.023)	(0.021)	(0.022)	(0.023)
HC	26.161 ***	26.796 ***	24.251 ***	
	(6.023)	(6.052)	(5.987)	
INFOR	0.093 ***	0.085 **		0.056 *
	(0.032)	(0.033)		(0.034)
TRA	0.031 **		0.044 ***	0.042 **
	(0.015)		(0.016)	(0.016)
FI		0.006	0.015 **	0.010
		(0.006)	(0.006)	(0.006)
_cons	0.122 ***	0.132 ***	0.120 ***	0.181 ***
	(0.017)	(0.016)	(0.017)	(0.009)
N	330.000	330.000	330.000	330.000
R^2^	0.547	0.542	0.543	0.523
R^2^_a	0.497	0.491	0.492	0.469

Note: Standard deviations are in parentheses, where ***, **, and * indicate significance at the 1%, 5%, and 10% levels, respectively.

## Data Availability

The data that support the findings of this study are available from the corresponding author, upon reasonable request.

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
