# Peer review of "Green Finance, Industrial Structure Upgrading, and High-Quality Economic Development–Intermediation Model Based on the Regulatory Role of Environmental Regulation"

_ijerph, 2023, doi:10.3390/ijerph20021420_

Round 1
Reviewer 1 Report
The main problems are as follows:
(1) The few local studies mentioned in this review do not conform to reality, which indicates that the literature tracking is insufficient.
(2) The most important basic work of this paper is to build the indicator system of high-quality economic development and green finance level. However, the indicator system of high-quality economic development is based on 5 of the 10 dimensions in the reference literature, and there are serious omissions in the selection of indicators.
(3) There are obvious defects in the design of the green financial measurement system. We only refer to the literature in 2014 and do not pay attention to the development of the latest green financial measurement. In particular, it is wrong to describe the carbon finance measurement with the proportion of carbon emissions in GDP. We should use carbon market trading, etc., and there are also insufficient considerations in other indicators such as green insurance.
(4) In the model design, why the panel model in this paper is not explained clearly.
Author Response
Dear reviewer, please see the attachment, thank you.

Reviewer 2 Report
The topic presented in this manuscript is of huge importance whether when it comes to green technology or green finance, so practically linking both topics is of great relevance to the world challenge nowadays and the coming future.
the introduction is good
part 2 is well dissected, i suggest to make the graphs more interesting to the reading especially that the topic is green so the authors can do better.
Part 3 holds a good analysis and the mediating variables presented along with the graphs and equations are of good interpretation.
I highly appreciate the work done in part 4 and have no addition to the models drawn by the authors in function of the different variables.
In conclusion , there is no doubt that the topic presented is of high interest for the Chinese economy as a model and for the rest of the middle and eastern ASian countries and the dissection of the conclusions one by one is a well done job for the authors.
Author Response

(The authors gave the same response as above.)

Reviewer 3 Report
This paper examines the impact of green finance on economic high-quality development and its transmission path using the intermediary model. I think this paper is an interesting and hot topic. But it needs a major revision before accepted. Some modifications should be conducted as follows:
1. There are several core variables involved in the paper,green finance ,industrial structure upgrading ,high-quality economic development, environmental regulation, I would suggest to further comb the relationship between the four variables.
2. The introduction is too long and lacks logic. The first paragraph has not cited any papers. It is suggested to edit the last part to focus on the research gap and article content (problems and reasons, methods and means, innovation and contributions).
3. The theoretical analysis section is too long, so it is suggested to be concise and more organized.
4. For Figure 3, I would suggest to add ordinate labels to distinguish high-quality economic development from green financial development.
5. Please confirm that "WAND database" is not "WIND database"(Line 422).
6. Add quotation to the part with formula in the text and explain the source.
7. The empirical analysis part can be related to the existing research theoretical results, and increase discussion and analysis.
8. The conclusion part tries to minimize the restatement of the empirical results, focusing on policy analysis, and connecting with the actual national conditions.
9. The references are generally old. It is suggested to reduce the number of long dated references and add new ones. In addition, the format of references is confusing.
10. The author needs to check the full text and pay attention to the expression, Such as lines 29, 109, and so on.
Author Response

(The authors gave the same response as above.)

Round 2
Reviewer 3 Report
The authors have made a comprehensive revision according to the review comments.I have no problem, I suggest receiving it.